# Precision Psychiatry for Obsessive-Compulsive Disorder: Clinical Applications of Deep Learning Architectures

**DOI:** 10.3390/jcm14072442

**Published:** 2025-04-03

**Authors:** Brian A. Zaboski, Lora Bednarek

**Affiliations:** 1Yale School of Medicine, Department of Psychiatry, Yale University, New Haven, CT 06510, USA; 2Department of Psychology, University of California, San Diego, CA 92093, USA; lbednarek@ucsd.edu

**Keywords:** obsessive-compulsive disorder, deep learning, neural networks, machine learning, biomarkers, precision psychiatry

## Abstract

Obsessive-compulsive disorder (OCD) is a complex psychiatric condition characterized by significant heterogeneity in symptomatology and treatment response. Advances in neuroimaging, EEG, and other multimodal datasets have created opportunities to identify biomarkers and predict outcomes, yet traditional statistical methods often fall short in analyzing such high-dimensional data. Deep learning (DL) offers powerful tools for addressing these challenges by leveraging architectures capable of classification, prediction, and data generation. This brief review provides an overview of five key DL architectures—feedforward neural networks, convolutional neural networks, recurrent neural networks, generative adversarial networks, and transformers—and their applications in OCD research and clinical practice. We highlight how these models have been used to identify the neural predictors of treatment response, diagnose and classify OCD, and advance precision psychiatry. We conclude by discussing the clinical implementation of DL, summarizing its advances and promises in OCD, and underscoring key challenges for the field.

## 1. Introduction

Obsessive-compulsive disorder (OCD) is characterized by intrusive thoughts and repetitive behaviors that significantly impair daily functioning [1,2]. It remains one of psychiatry’s most complex challenges, with up to 50% of patients failing to achieve remission despite gold-standard interventions such as exposure and response prevention (ERP) and serotonin reuptake inhibitors (SSRIs) [3,4]. Current barriers to care include misdiagnosis [1], heterogeneous symptom presentations [2,3], family accommodation [4], high comorbidities with depression and anxiety disorders [5], ERP dropout [6], and limited accessibility to specialized care [7,8]. Moreover, challenges in pharmacological intervention include increased side effect burden at OCD-specific dosages [9], treatment resistance, and limited research on possible augmentation strategies (e.g., brexpiprazole and cariprazine) [10,11]. Because international treatment recommendations provide little to no guidance as to which patients are most likely to improve or to be treatment refractory [12,13,14], there is a critical need for computational approaches that can analyze complex patient data to improve diagnostics and inform personalized interventions.

Deep learning, a subset of artificial intelligence (AI), offers the analytic tools to understand and predict psychiatric illnesses, and its wide-ranging use in psychiatry can provide OCD researchers with an implementation model [15]. One application is in analyzing neuroimaging data. For example, in schizophrenia, AI-enhanced analyses have identified disruptions in the prefrontal cortex, hippocampus, and default mode network [15]. Through superior pattern recognition, these analyses found disrupted white matter tracts in the corpus callosum, cingulum bundle, and frontotemporal fasciculi. Another investigation [16] used fMRI-based frontoparietal activations to predict clinical improvement in *N* = 82 individuals with early psychosis. Deep learning outperformed machine learning in predicting both binary improvement status (70% accuracy) and continuous change in psychiatric symptoms, with left dorsolateral prefrontal cortex activation identified as a significant predictor. Additionally, numerous large-scale, AI-driven projects are underway, including the Master Data Platform (MDP), an AI-driven project utilizing machine and deep learning for the personalized treatment of eating disorders in Italy [17]. The MDP aims to enhance diagnosis, treatment planning, and relapse prediction through data analysis and a patient-facing chatbot, improving care accessibility, personalizing treatments, and optimizing treatment pathways within the Italian mental healthcare system.

Deep learning has also been utilized in pharmacogenetics studies, which integrate multimodal data types—including single nucleotide polymorphisms (SNPs), DNA methylation data, gene expression data, and phenotypic data (demographic and clinical information)—to optimize treatment response prediction. These approaches provide another promising application for OCD research. For instance, in major depressive disorder (MDD), integrating multimodal information yielded high performance in pharmacotherapy prediction. A deep neural network predicted antidepressant treatment response and remission, with better prediction than a traditional logistic regression [18]. Likewise, in a study with children diagnosed with attention-deficit/hyperactivity disorder (ADHD), researchers identified genetic variants associated with pharmacological treatment response using a genome-wide association study (GWAS) and developed deep learning models to predict this response based on genomic data [19]. They identified two significant genetic loci through the GWAS (TMEM117 and MYO5B) that were associated with pharmacological treatment response. Their model, which was designed for image identification, also demonstrated promising predictive performance for treatment response, highlighting NKAIN2 as a key contributing gene.

These applications illustrate deep learning’s utility in psychiatry for prediction, pattern recognition, and patient care, while providing a basis for personalized treatment approaches. Although the field of OCD has been using machine learning for at least a decade [20], it has been slow to adopt deep learning methods. To introduce readers to deep learning, this paper uniquely focuses on specific deep learning models (i.e., architectures) and their applications in OCD, providing clinicians and researchers with an accessible and practical understanding of how these techniques can transform patient care. In this focused review, we highlight applications such as advanced prediction, image-based analysis, real-time symptom monitoring, data generation, and dynamic therapeutic support. The representative applications discussed in this review are selected based on clinical relevance, methodological innovation, and the potential for translation to clinical practice. Ultimately, we aim to equip researchers and practitioners with basic knowledge of how some fundamental architectures work, offering clear applications for the field and evidence-based frameworks for adopting deep learning tools while addressing ethical challenges. For a more technical introduction with broader implications in psychiatry or specific reference to neuroimaging, we refer readers elsewhere [21,22,23,24,25,26].

## 2. Predictive Modeling and Biomarker Identification with Feedforward Neural Networks

Feedforward neural networks (FFNNs) are the simplest deep learning architecture. In a FFNN, information flows unidirectionally from an input layer, then through one or more hidden layers, and finally to an output layer. Each neuron in a layer is connected to neurons in the subsequent layer through weights that determine the strength of their relationships. The input data are transferred to hidden layers, which process the data and transfer it to the output layer. The output layer then provides the results of the network’s computations (e.g., outputs based on the input data; Figure 1). The backpropagation algorithm, which adjusts weights to minimize error and optimize performance, then propagates prediction errors backward through the network to refine its weights. Nonlinear relationships can be included with functions included in the hidden layer or output layers of the architecture.

This simple architecture holds substantial promise for advancing personalized approaches in OCD diagnosis and treatment. FFNNs have been instrumental in predicting treatment outcomes for patients undergoing ERP or SSRIs, with performance surpassing traditional models such as logistic regression in accounting for complex interactions among clinical, epidemiological, and neuropsychological variables [27]. For instance, using baseline quantitative EEG (qEEG) data, researchers achieved 80% accuracy in predicting response to transcranial magnetic stimulation (TMS), a neuromodulation technique gaining traction as a treatment for refractory OCD, thus highlighting the role of FFNNs in identifying neural biomarkers that guide treatment personalization [28]. Such applications provide an opportunity to refine treatment protocols by matching patients to interventions likely to yield optimal outcomes.

The flexibility of FFNNs also makes them well-suited for capturing the heterogeneity of OCD presentations across different populations. In one study, a FFNN successfully predicted OCD severity based on personality traits, religiosity, and demographic factors, offering insights into culturally sensitive symptom presentations such as heightened religious obsessions in Catholic-majority groups [29]. These findings illuminate how FFNNs can enhance clinical assessment by tailoring diagnostic and therapeutic strategies to patients’ cultural and individual contexts. Moreover, by integrating multimodal data sources, such as neuroimaging, clinical profiles, and ecological momentary assessments, FFNNs hold the potential to dynamically monitor symptoms and predict relapse risk in real-world settings, paving the way for more adaptive and context-aware interventions.

## 3. Image-Based Classification: Convolutional Neural Networks

Convolutional neural networks (CNNs) are among the most widely used deep learning architectures for analyzing unstructured data and are commonly encountered in psychiatric research [30]. In particular, CNNs excel at extracting spatial hierarchies of features from image data, making them well-suited for neuroimaging tasks [23]. A schematic of the LeNet architecture, one of the earliest CNNs, is shown in Figure 2 [31]. In CNNs, images are processed through layers of convolutional filters that detect patterns such as edges or textures. Pooling layers then reduce the spatial dimensions of these feature maps, simplifying the data while retaining critical information. Additional convolutional and pooling layers can be added to extract increasingly complex features, culminating in fully connected layers that produce predictions or classifications [32].

CNNs have been successfully applied to classify OCD patients and predict treatment outcomes using neuroimaging data, facilitating personalized care strategies. For instance, Kalmady et al. (2022) [33] applied resting-state functional magnetic resonance imaging (fMRI) to distinguish 188 OCD patients from 200 healthy controls with high accuracy (80%) and sensitivity (83%), demonstrating a CNN’s potential to identify neural biomarkers associated with OCD. Such insights could guide treatment selection, including neuromodulation techniques such as TMS. Beyond diagnosis, CNNs enable the development of AI-driven predictive tools that reduce reliance on manual feature selection, accelerating the analysis of complex neuroimaging datasets [34]. These advances create opportunities to translate research findings into real-world clinical workflows, such as integrating neurobiological predictors into decision-making systems for treatment-resistant OCD.

Moreover, CNNs are being explored for transdiagnostic features relevant to OCD, opening new avenues for early intervention and risk assessment. For example, CNN-based models have been used to identify subtle physiological markers, such as facial cues correlated with suicidal ideation, achieving 95% accuracy [35]. These models leverage explainable AI techniques to provide clinicians with interpretable outputs, detailing the neural or behavioral features driving predictions. The potential to combine CNN-derived biomarkers with patient history and symptom profiles not only enhances diagnostic precision but also equips clinicians with actionable data to inform therapeutic interventions.

## 4. Temporal Tracking of Symptoms: Recurrent Neural Networks

Recurrent neural networks (RNNs) are designed to analyze sequential data by incorporating dependencies between time points, making them particularly suited for time-series data such as fMRI, EEG, and ecological momentary assessments (EMAs). Unlike FFNNs or CNNs, RNNs “remember” prior inputs by using feedback loops that pass information from one time step to the next [36]. This enables RNNs to model dynamic systems where the current state depends on previous states (see Figure 3).

RNNs have demonstrated significant potential in OCD research, particularly for real-time symptom monitoring, predicting treatment responses, and optimizing dynamic therapeutic interventions. Long short-term memory (LSTM) networks, an advanced RNN variant, mitigate limitations such as vanishing gradients, enabling the modeling of long-term dependencies in sequential data. For instance, Mishra et al. (2025) [37] utilized LSTMs to analyze EEG signals and classify emotional states (positive, negative, neutral) with 97% accuracy, providing a pathway for detecting anxiety-related patterns that frequently co-occur with OCD. In a novel application, Kirsten et al. (2021) [38] developed an unobtrusive system to detect compulsive behaviors in OCD using wearable sensor data and personalized federated learning algorithms. The system achieved an area under the receiver operating characteristic curve (AUC) of 0.954, enabling accurate identification of repetitive compulsions such as excessive handwashing. By combining these tools with dynamic assessment frameworks such as EMA, clinicians could better differentiate between adaptive behaviors and symptomatic compulsions, identify early signs of symptom exacerbation, and intervene proactively with tailored therapeutic strategies.

## 5. Data Generation and Augmentation: Generative Adversarial Networks

Generative adversarial networks (GANs) generate new data based on existing input, such as text, images, or sounds. GANs consist of two neural networks—the generator and the discriminator—that work in opposition to improve their respective tasks. The generator creates synthetic data resembling the training examples, while the discriminator evaluates whether the generated data are real or fake. Through iterative training, both networks refine their abilities until the generator produces realistic outputs indistinguishable from the original data (Figure 4) [32].

Based on prior research in psychiatry, GANs hold significant potential for advancing OCD research and treatment, particularly in generating clinical insights from complex datasets or addressing unmet needs in patient populations. While no studies have explicitly examined GAN applications in OCD, their success in related fields provides a roadmap for future implementation. For example, GANs have been used to decompose neuroimaging data into components relevant and irrelevant to disease progression in Alzheimer’s research, enabling the identification of biomarkers tied to cognitive decline [39]. Similarly, GANs have augmented training datasets by generating synthetic medical images, such as imputing missing PET data based on corresponding MRI scans, thereby improving diagnostic accuracy [40]. Applying these techniques to OCD could include imputing missing data in longitudinal Y-BOCS scores or using neuroimaging modalities to predict illness trajectories and treatment responses. By synthesizing datasets that integrate diverse imaging, behavioral data, or neurophysiological data, GANs could accelerate the development of personalized interventions targeting unique OCD presentations.

GANs also offer promising solutions to challenges pervasive in OCD research, such as addressing demographic disparities and reducing the costs of large-scale data collection. Conditional GANs, for instance, have successfully generated virtual patient cohorts to model clinical biomarker distributions across underrepresented racial and ethnic groups, ensuring more inclusive and equitable datasets for diabetes research [41]. In the context of OCD, GANs could simulate culturally and demographically diverse patient profiles, enabling the development of interventions tailored to specific populations and reducing biases in treatment recommendations. Moreover, by generating synthetic data to simulate the effects of new therapies, GANs could aid in preclinical testing and refine intervention strategies before costly clinical trials. These efforts not only improve the generalizability of research findings but also bridge the translational gap between experimental models and real-world applications.

## 6. Automated Screening and Dynamic Therapeutic Support: Transformers

Transformers have revolutionized AI by replacing sequential processing with self-attention mechanisms, enabling models to analyze entire datasets simultaneously while identifying contextual relationships across data points [42]. Unlike RNNs, which process data step-by-step, transformers weigh the importance of each input element (e.g., words in a sentence, pixels in an image) relative to others, allowing them to capture long-range dependencies and global patterns. A transformer consists of an encoder-decoder structure, where the encoder maps input data into a high-dimensional representation, and the decoder generates output based on that representation. The core innovation is the multi-head attention layer, which computes the interactions between all the input elements in parallel [42]. This architecture underpins modern large language models (LLMs) such as BERT and GPT, which excel at tasks ranging from text generation to multimodal data integration [43,44].

Transformers are beginning to demonstrate promising applications in OCD diagnosis and treatment personalization, particularly through the use of LLMs and multimodal data integration. In a diagnostic study, Kim et al. (2024) [45] compared the performance of three LLMs—ChatGPT-4 (OpenAI, Inc., San Francisco, CA, USA, April 2023 version), Gemini Pro (Google Inc., Mountain View, CA, USA, April 2024 version), and Llama 3 (Meta Inc., Menlo Park, CA, USA, April 2024 version)—to medical and mental health professionals in identifying OCD across case vignettes. With ChatGPT-4 achieving perfect diagnostic performance (100%), these architectures surpassed human practitioners in accuracy, highlighting their potential as decision-support tools in psychiatric practice. Another study evaluated ChatGPT-3.5, ChatGPT-4, Claude, and Gemini (no version numbers specified) using client vignettes representing diverse OCD subtypes, demonstrating that LLMs consistently outperformed psychotherapists in recognizing OCD (90–100% vs. 47.3% accuracy) and recommending evidence-based treatments [46]. These findings underscore the opportunities for transformers to augment clinical workflows by accurately identifying OCD presentations, minimizing diagnostic delays, and tailoring interventions to specific symptom subtypes. Furthermore, explainable AI tools integrated into these models help clinicians interpret predictions and understand differential recommendations for personalized treatment pathways.

## 7. Evaluating Model Performance

### 7.1. Clinical Evaluation

The clinical utility of deep learning models in OCD critically depends on rigorous clinical validation, which ensures patient benefit and safety before integrating AI technologies into practice [47]. Clinical validation requires independent external datasets collected in newly recruited patients or at different sites than the dataset used for algorithm development. These datasets should effectively represent the target patients undergoing a given diagnostic or predictive procedure in clinical practice settings to obtain an unbiased assessment of an AI system [47]. For example, in OCD studies, models are increasingly tested on different OCD symptom dimensions (e.g., contamination versus checking) [38], performance is stratified by common comorbid conditions [28], and studies routinely report performance across demographics (e.g., age, gender, ethnicity) [29]. However, because AI research in OCD is in its infancy, models are not yet clinically validated on external datasets. Further accounting for this heterogeneity is a crucial next step for deep learning to become clinically actionable in the real world, where symptom presentations are more heterogeneous.

### 7.2. Model Evaluation

In AI and machine learning, validation is a technical term that refers to a specific step in algorithm development. For example, a large dataset is used to train a model, then validation involves fine-tuning it. Model performance is then tested on unseen data from the same dataset (a test set) [32]. To facilitate the process of model validation, studies frequently employ cross-validation, where data are systematically divided into training and testing sets. For example, imagine splitting a dataset into smaller groups, then repeatedly training a model on most groups and testing it on the leftover one. By rotating which group is used for testing, researchers can obtain a reliable measure of performance. Because training, testing, and validation use the same data, an external dataset is crucial for clinical validation [48,49].

When evaluating models, researchers must also consider overfitting and underfitting. In high-variance situations (e.g., the presence of irrelevant or extraneous information, sampling errors, or measurement inaccuracies), overfitting can occur when models memorize noise or random variations in the training data. Conversely, simplistic models can lead to underfitting (high bias), where they fail to capture underlying patterns in the data. Overfitting is closely tied to this bias–variance tradeoff; high variance often leads to overfitting, while high bias results in underfitting (expert readers may note that bias-variance tradeoffs behave somewhat unpredictably in the context of neural networks [50,51,52]). Kalmady et al. used a five-fold cross-validation approach in their CNN-based neuroimaging study to avoid overfitting [33]. Similarly, Metin et al. [28] managed their clinical dataset with leave-one-out cross-validation, a technique particularly valuable for the smaller datasets in most OCD research.

There are several metrics for evaluating how well a model performs at a task. These include sensitivity (correctly identifying true OCD cases), specificity (correctly identifying non-OCD cases), and overall accuracy. For classification tasks such as diagnostic prediction, the area under the receiver operating characteristic curve (AUC) provides a comprehensive assessment of a model’s ability to discriminate between classes (e.g., responder to OCD treatment vs. non-responder) [53]. Values range from 0.5 (no discrimination ability) to 1.0 (perfect discrimination). For instance, the paper by Kirsten et al. (2021) [38] for detecting compulsive behaviors achieved an AUC of 0.954, indicating excellent discriminative ability. For a more thorough treatment of model performance metrics, see Naidu et al. (2003) [53].

## 8. AI-Driven Personalization: Clinical Implementation

The integration of deep learning into OCD research and clinical care marks a paradigm shift from one-size-fits-all interventions to precision psychiatry. By leveraging deep learning architectures and multimodal data, AI enables dynamic, individualized treatment strategies that adapt to patient-specific symptoms, biomarkers, and environmental contexts. Below, we highlight key themes underlying this transformation.

### 8.1. Biomarker Identification

Advances in neuroimaging, EEG, and molecular biomarkers have enabled AI models to predict treatment responses in OCD with increasing precision. For example, EEG complexity metrics, such as approximate entropy in the beta frequency band, have been shown to differentiate treatment-resistant from treatment-responsive patients with 89.66% accuracy [54]. Similarly, neuroimaging studies using fMRI have identified distinct patterns of cortical-striatal-thalamic-cortical (CSTC) circuitry activity that correlate with OCD symptom severity and treatment outcomes [55]. These biomarkers are now being integrated into AI-driven models to predict responses to interventions such as TMS. For example, one study achieved 80% accuracy in predicting treatment response by analyzing connectivity patterns in CSTC circuits using CNNs and machine learning algorithms [33]. In another application, EEG theta-band power predicted TMS outcomes with 80% accuracy, enabling clinicians to prioritize neuromodulation for eligible patients [28]. These methods vastly improve upon studies that primarily investigate clinical correlates (e.g., EEG) [56].

Combining theory with the models and data sources discussed in this review, the first AI-created drug was developed—for OCD [57]. Compound DSP-1181 [58] progressed from initial screening to preclinical testing in under 12 months—a stark contrast to the industry average of 4–6 years [59]. By leveraging AI’s ability to analyze vast chemical libraries and optimize molecular structures, researchers aim to accelerate the identification of promising candidates while reducing costs and increasing efficiency. Of course, challenges remain, as only 10% of compounds entering Phase 1 trials typically achieve regulatory approval [59]. Nevertheless, AI’s capability to identify novel targets and optimize animal models for preclinical studies holds promise for enhancing success rates in OCD drug development [40].

### 8.2. Digital Therapeutics and Real-Time Interventions

Deep learning architectures represent a transformative leap in mental health research and clinical care. These technologies are poised to address long-standing challenges in psychiatry, such as the need for scalable, personalized interventions, and a nuanced understanding of complex mental health conditions such as OCD. These models can synthesize patient-specific data, interpret unstructured inputs (e.g., therapy transcripts or self-reports), and provide actionable insights for diagnosis and treatment. Their ability to integrate multimodal datasets, including behavioral, physiological, and linguistic data, heralds a new era in precision psychiatry, offering the potential to optimize treatment strategies and improve accessibility to evidence-based care. In the context of OCD, these architectures are already becoming integrated into the psychotherapy delivery process. AI-powered mobile apps such as Wysa and Woebot deliver personalized ERP by adjusting task difficulty with FFNNs and RNNs based on real-time symptoms and physiological data (e.g., galvanic skin response). Although more research is needed on their long-term benefit—and optimization is required to improve the user experience—many provide effective mental health support and successfully meet a high need for services (Farzan et al., 2025) [60].

Moreover, transformers such as ChatGPT are reshaping therapeutic workflows. Not only are they proving themselves diagnostically [45], they are being tailored to provide psychoeducation to each patient’s level of understanding, identify cognitive distortions in text, challenge automatic thoughts, create reflection prompts for homework assignments, and monitor CBT treatment fidelity [61]. Another approach is the development of AI-guided exposure therapy with virtual reality environments and an AI therapist [62]. In this study, the AI therapist, driven by GPT-4, interacted with patients, guiding them through exposure exercises for the fear of heights while monitoring their emotional state and adjusting the intensity. The system’s design incorporated ethical principles translated into technical requirements, including trust-building and bias detection. Preliminary evaluation showed that patients were satisfied with the virtual reality environment but noted that the AI therapist needed improvements in human-like interaction and empathy.

## 9. Challenges and Future Directions

### 9.1. General Limitations of AI in OCD

#### 9.1.1. Uncertain Comparative Effectiveness

Although deep learning models show promising results in predicting treatment outcomes and identifying biomarkers for OCD, we do not yet know how effective these models are compared to traditional psychiatric assessment tools. The reported performance metrics, while impressive in research settings, lack validation against validated clinical assessments in real-world psychiatric practice. For example, although vignettes have been promising in clinical prediction [45,46], they do not reflect the complex, multi-step assessment process required for real-world clinical decisions. A curated dataset with 2400 real patient cases with common abdominal pathologies demonstrates this point; in this study, LLMs gathered and synthesized additional information to reach a diagnosis and treatment plan. It was found that clinicians had a mean diagnostic accuracy of 88%, while the LLM’s mean accuracy ranged from 46–54%, with them failing to accurately diagnose patients across all pathologies, follow diagnostic or treatment guidelines, or correctly interpret lab results. Consequently, researchers interested in applying LLMs in OCD research should consider testing them on real-world data (e.g., intakes, therapy transcripts). Future research must directly compare deep learning approaches with established clinical instruments such as the Y-BOCS, standardized diagnostic interviews, and clinical judgment through prospective studies with diverse patient populations.

#### 9.1.2. Cost and Resource Constraints

Implementation in clinical settings faces significant barriers related to cost and resource requirements. The development, training, and deployment of neural networks require substantial computational infrastructure, including high-performance hardware, specialized software, and technical expertise not readily available in most mental health settings. The financial investment needed for these resources poses challenges for healthcare systems already operating under budget constraints. Additionally, the ongoing maintenance, model updates, and technical support needed for AI systems add further operational costs. These resource limitations are particularly problematic for community mental health centers and rural practices, potentially exacerbating existing disparities in access to innovative treatments for OCD patients. However, as AI research is in its early stages, with most studies being proof-of-concept or model validation [63], the cost for clinical deployment is too early to determine.

#### 9.1.3. Limited Patient Perspectives Data

A critical research gap is the lack of data on how patients with OCD perceive AI-driven interventions. Understanding patient perspectives is essential for designing acceptable and engaging AI-based interventions, as technology adoption ultimately depends on patient willingness to incorporate these tools into their treatment. In a review of AI applications in clinical practice, 3/51 studies examined how patients evaluated AI applications, and all of them reported positive results [63]. For example, when evaluating an AI-based diabetic retinopathy screening tool, researchers found that 96% of screened patients were satisfied with the AI tool and 78% of patients in the follow-up survey preferred AI screening over manual screening [64]. At the same time, patient attitudes toward algorithmic decision-making, concerns about data privacy, and preferences regarding the balance of human and AI interaction may differ for OCD patients, who may have different perspectives of trust, control, and certainty [65,66,67]. Future studies should systematically assess patient experiences, concerns, and satisfaction with deep learning applications in OCD care through qualitative and quantitative methods.

#### 9.1.4. Reproducibility

The reproducibility of deep learning models represents a significant limitation in current OCD research. Many studies utilize small, homogeneous samples with unique preprocessing pipelines and model architectures, making replication difficult. The complexity of neural networks further complicates reproducibility, as the internal mechanisms driving predictions often remain difficult to interpret. Additionally, differences in data collection methods, clinical assessment procedures, and outcome definitions across research sites introduce variability that affects model performance. Addressing these challenges requires the standardized reporting of methodologies, open-source code sharing, public dataset availability (while maintaining privacy), and multi-site validation studies. In addition, explainable AI techniques can be used to reveal the decision-making process of a model [35].

#### 9.1.5. Interpretability

Traditional methods for characterizing outcomes in OCD have used tools such as ANCOVA or mixed models (e.g., to model a response to an SSRI in a randomized clinical trial). A downside to these linear models is that they can only approximate the high heterogeneity and multidimensionality inherent to OCD. Yet, these approximated models provide exact interpretations that provide researchers with insight into underlying mechanisms. For example, a researcher can examine the estimates provided by their statistics and conclude that an SSRI is probably effective. When a researcher is interested in mechanisms, such analyses may be appropriate.

By contrast, for questions related to prediction, these models may miss complex, nonlinear relationships, sacrificing predictive accuracy for interpretability [68]. Machine and deep learning approaches alleviate this problem by using models that fit the data exactly. As discussed in this review, these incorporate complex, nonlinear relationships that can generalize well to unseen data, which improves overall prediction. However, these strengths come at the expense of interpretability, providing only approximate interpretations of underlying mechanisms.

This tradeoff between methods that are adept at understanding mechanisms and prediction creates a clinical conundrum: clinicians need to understand the rationale behind their recommendations, yet deep learning models require clinicians to trust predictions without fully understanding how they arrive at their output. This is a problem that connects to clinical validation (discussed above) and showcases why these models need to face greater scrutiny than typical clinical applications. This scrutiny involves incorporating experts throughout model development and clinical practice implementation [69]. Clinicians may also need specialized training to interpret and appropriately act on model outputs to avoid their misapplication or the rejection of potentially valuable outputs due to discomfort with the model’s opaque nature. Applications of deep learning in OCD are young, and interpretation guidelines are not yet well-formulated. We encourage clinicians to continue their professional development and join organizations that give them a decision-making voice (e.g., the International OCD Foundation’s Special Interest Group on Artificial Intelligence) [70].

### 9.2. Ethical Limitations

#### 9.2.1. OCD Research

AI application in OCD grapples with significant ethical considerations, particularly concerning data privacy, security, and potential biases [71,72]. The collection and utilization of personal data, especially sensitive psychiatric information, necessitate stringent measures to ensure data protection and confidentiality. Researchers must carefully navigate data access protocols and implement robust security measures to prevent unauthorized disclosure or misuse [62]. Furthermore, algorithmic bias poses a substantial challenge, as LLMs can inadvertently perpetuate harmful stereotypes present in their training data, leading to inequitable or discriminatory outcomes. To mitigate these risks, researchers must prioritize cultural sensitivity, inclusivity, and fairness in AI system design and evaluation. This involves curating diverse and representative datasets, employing bias-detection techniques, and continuously monitoring performance across different demographic groups to prevent exacerbating health disparities. Moreover, transparency in AI development is essential for fostering trust and accountability. Researchers should provide clear and comprehensive information about the system’s development team, ownership, funding sources, business model, training methodologies, and primary beneficiaries [71].

In addition, assessing AI’s efficacy and potential harm in research settings raises complex ethical dilemmas. The rapid proliferation of digital platforms necessitates a thorough and ongoing evaluation of their potential benefits and risks [71]. Without such an evaluation, timely responses to unforeseen adverse effects or unintended consequences become exceedingly difficult [62]. To address this gap, there is a growing need for structured frameworks and ethical guidelines tailored to evaluate LLM tools in mental health research. These frameworks should encompass key ethical principles, such as beneficence, non-maleficence, respect for autonomy, and justice [71]. The inherent complexity of AI interactions underscores the importance of actively involving users in the development process and continuously monitoring AI performance to detect and address any unintended consequences that may arise [62].

#### 9.2.2. OCD Clinical Care

In the realm of clinical care, AI tools designed for OCD treatment, such as conversational agents and virtual reality exposure therapy (VRET), introduce concerns about the distortion of the therapeutic frame, the potential blurring of boundaries, and the compromise of the patient–therapist relationship [62,73]. While AI-driven interventions may simulate therapeutic conversations, they often lack the depth and nuance of genuine engagement with a human therapist, possibly endangering user autonomy and psychological integrity if not properly understood. Moreover, on-demand conversational access may undermine established therapeutic practices, hindering opportunities to cultivate realistic interpersonal relationships and support networks [73]. There are also ethical considerations with the use of autonomous VRET, and ensuring minimum standards concerning autonomy, control, fairness, transparency, reliability, security, and data protection [62].

Furthermore, ensuring patient safety and effectively managing crisis situations are paramount ethical considerations in AI-driven OCD care [72]. For instance, AI systems must be equipped to detect signs of suicidality, recognize contraindications for specific interventions, and proactively manage symptom deterioration [62]. Clear protocols and readily accessible resources for handling emergencies are essential, as are seamless mechanisms for incorporating human therapists into the treatment process when necessary [71]. The transparency and comprehensibility of AI algorithms are equally critical to empower patients to understand how these tools function and prevent potential misuse [62]. Addressing the digital divide and ensuring equitable access to AI-based interventions are also vital for responsible and ethical implementation in clinical settings, preventing further disparities.

## 10. Toward Precision OCD Care

Despite the barriers inherent to a new and rapidly growing field, researchers are uncovering novel biomarkers, predicting treatment responses, and tailoring interventions to individuals. AI-driven tools are enabling the identification of neural predictors of treatment outcomes, the generation of personalized exposure stimuli, and the development of dynamic therapeutic platforms that adapt to real-time patient feedback. Figure 5 summarizes the types of inputs, architectures, clinical applications, ethical considerations, and outcomes discussed in this focused review.

Although this review discusses specific applications for each architecture, Figure 5 shows that deep learning is exceedingly flexible and many different models can process varying types of input data. As Figure 5 also shows, ethical considerations including data privacy, algorithmic bias, and equitable access to AI-driven interventions are key considerations for all stages of model development and deployment. Moreover, results require constant iteration for improvement, perhaps requiring different data or alternative architectures that may give rise to new challenges. As these methods mature, taking a multidisciplinary approach that combines cutting-edge AI with rigorous clinical validation and ethical oversight will be essential to realize their potential in clinical care.

## Figures and Tables

**Figure 1 jcm-14-02442-f001:**
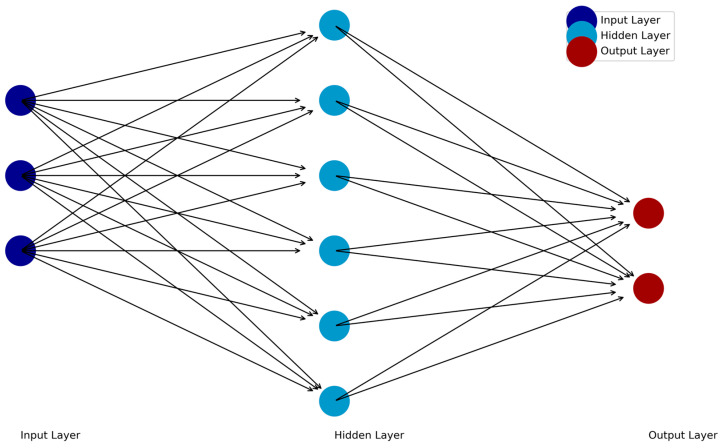
Schematic of the feedforward neural network (FFNN).

**Figure 2 jcm-14-02442-f002:**
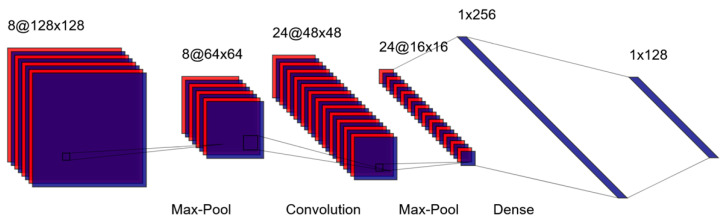
Schematic of the LeNet convolutional neural network. Purple areas represent the actual feature maps or activation values at each layer of the network. These show the data as it is being processed. The red outlines or borders highlight the boundaries of each feature map or filter. These red borders help visualize the distinct channels or filters at each layer, making it easier to see the three-dimensional structure of the data as it moves through the network. Lines indicate which parts of one layer connect to the next layer.

**Figure 3 jcm-14-02442-f003:**
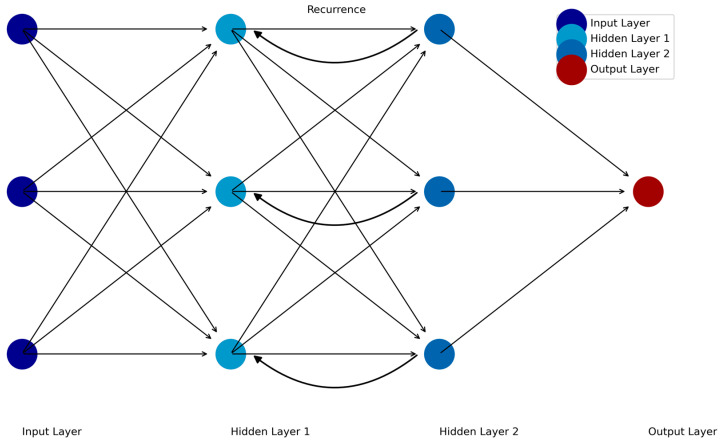
Schematic of the recurrent feedforward neural network.

**Figure 4 jcm-14-02442-f004:**
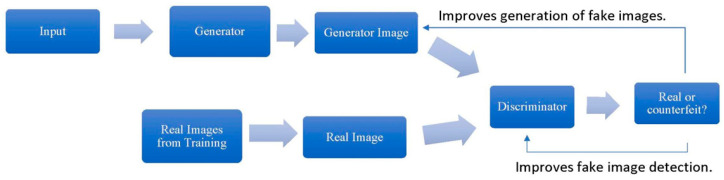
General procedure for training a generator and discriminator.

**Figure 5 jcm-14-02442-f005:**
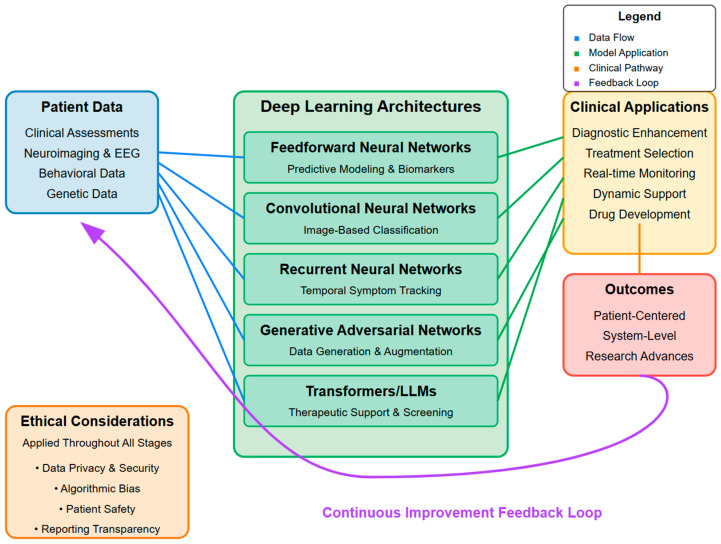
Flowchart summarizing deep learning inputs, models, outcomes, applications, and ethical considerations.

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
