# Peer review of "Precision Psychiatry for Obsessive-Compulsive Disorder: Clinical Applications of Deep Learning Architectures"

_jcm, 2025, doi:10.3390/jcm14072442_

Round 1

Reviewer 1 Report

Comments and Suggestions for Authors

This review explores the role of deep learning (DL) in precision psychiatry for obsessive-compulsive disorder (OCD). It introduces five key DL architectures—feedforward neural networks, convolutional neural networks, recurrent neural networks, generative adversarial networks, and transformers—and their applications in diagnosis, treatment response prediction, and biomarker identification. The study highlights neuroimaging, EEG, and AI-driven clinical tools as transformative in OCD care. It discusses real-world implementation challenges, including algorithmic bias, ethical concerns, and data privacy issues. The review concludes by outlining future directions for AI integration in psychiatric research and treatment.

Introduction

The novelty of the review is not well articulated—how does it differ from previous studies on AI in psychiatry?

Lacks discussion on the current state of DL adoption in clinical psychiatry—what are the existing applications in OCD treatment?

The introduction on OCD and its clinical challenges is too concise and requires more depth. Please provide adding references that can deepen the discussion also on the issue of  pharmacologicalresistance and its treatment and on how AI can be useful in this situations. To expand AI applications in other psychiatry fields the following references may be appropriate (Di Stefano V, et al.,Decoding Schizophrenia: How AI-Enhanced fMRI Unlocks New Pathways for Precision Psychiatry. Brain Sci. 2024 Nov 27;14(12):1196.) – (Monaco F, et al., An advanced Artificial Intelligence platform for a personalised treatment of Eating Disorders. Front Psychiatry. 2024 Aug 6;15:1414439. )

To expand the introduction about treatment resistant OCD therapeutic approaches cite this one recent paper (Martiadis V, Serotonin reuptake inhibitors augmentation with cariprazine in patients with treatment-resistant obsessive-compulsive disorder: a retrospective observational study. CNS Spectr. 2024 May 22:1-4. doi: 10.1017/S1092852924000348)

The transition to deep learning applications is abrupt—a smoother introduction linking computational psychiatry to precision medicine would improve clarity.

Methods

No systematic review methodology—was a structured literature search conducted, or were studies selected based on expert opinion?

Lack of inclusion/exclusion criteria—what were the criteria for choosing the reviewed studies?

No mention of validation techniques—how were the deep learning models assessed for accuracy and bias?

Results

Limited quantitative analysis—how effective are these DL models compared to traditional psychiatric assessment tools?

No comparison of model performance—which architecture (e.g., CNN vs. RNN) performs best for specific tasks?

Few real-world clinical validation examples—have any of these models been tested in hospitals or psychiatric clinics?

Discussion

No discussion on interpretability—how can clinicians trust and understand AI-generated predictions?

Fails to address cost and resource constraints—can psychiatric clinics feasibly integrate these tools?

Overlooks patient perspectives—how do OCD patients perceive AI-driven interventions?

Comments on the Quality of English Language

Some minor grammar and style mprovements

Line 27-29:
“OCD remains one of psychiatry’s most complex challenges, with up to 50% of patients failing to achieve remission despite gold-standard interventions like exposure and response prevention (ERP) and serotonin reuptake inhibitors (SSRIs).”
Revision:
“OCD presents a significant challenge in psychiatry, with up to 50% of patients not achieving remission despite first-line treatments like exposure-response prevention (ERP) and SSRIs.”

Line 97-99:
“Recurrent Neural Networks (RNNs) are designed to analyze sequential data by incorporating dependencies between time points, making them particularly suited for time-series data such as fMRI, EEG, and ecological momentary assessments (EMA).”
Revision:
“Recurrent Neural Networks (RNNs) specialize in processing sequential data, making them ideal for time-series applications like fMRI, EEG, and ecological momentary assessments (EMA).”

Line 186-188:
“The first AI-created drug was developed—for OCD. Compound DSP-1181 progressed from initial screening to preclinical testing in under 12 months—a stark contrast to the industry average of 4–6 years.”
Revision:
“The first AI-designed drug for OCD, DSP-1181, advanced from initial screening to preclinical trials in under 12 months—significantly faster than the industry norm of 4–6 years.”

Line 240-242:
“Ensuring patient safety and effectively managing crisis situations are paramount ethical considerations in AI-driven OCD care.”
Revision:
“Patient safety and crisis management are critical ethical concerns in AI-driven OCD treatment.”

Reviewer 2 Report

Comments and Suggestions for Authors

This review explores the applications of deep learning (DL) in precision psychiatry for obsessive-compulsive disorder (OCD). The authors discuss various DL architectures, including feedforward neural networks, convolutional neural networks, recurrent neural networks, generative adversarial networks, and transformers, and their role in biomarker identification, symptom classification, and treatment prediction. The paper highlights key clinical applications, such as predicting treatment response to exposure and response prevention (ERP), serotonin reuptake inhibitors (SSRIs), and transcranial magnetic stimulation (TMS). The authors argue that integrating AI into psychiatric research can enhance diagnosis, personalize treatment, and improve clinical outcomes. However, they also acknowledge challenges related to ethical considerations, algorithmic bias, and clinical implementation.

The introduction should better define the main goal of the review. Is it to propose a new DL-based framework for OCD treatment, or simply to summarize existing applications? A clearer research question would strengthen the focus.

While the potential of DL is well described, the discussion of its limitations (e.g., overfitting, interpretability, reproducibility issues) is superficial. A deeper analysis of the challenges and risks would improve academic rigor.

When talking about ocd challenges mention also treatment difficulties and new approaches such as augmentation therapies. In this context cite the recent paper by Martiadis et al., 2024  (Martiadis V, Brexpiprazole Augmentation in Treatment Resistant OCD: Safety and Efficacy in an Italian Sample. Psychiatr Danub. 2024 Sep;36(Suppl 2):396-401. ) dealing for the first time with brexpiprazole augmentation.

A flowchart illustrating the DL-driven OCD diagnosis and treatment workflow would benefit non-expert readers.

To strengthen the discussion, consider integrating the following references:

Gillan CM, Kostro D, Whelan R, et al. "Functional neuroimaging and machine learning predict individual vulnerability to compulsive behaviors." Neuron. 2016; 92(3): 517-527.

 Demonstrates how machine learning models predict OCD vulnerability.

The discussion on model validation is limited. The authors should emphasize the importance of cross-validation and external validation in ensuring generalizability.

The paper does not discuss bias-variance trade-off issues in DL models, which could impact clinical reliability. Please briefly argument about it

Round 2

Reviewer 1 Report

Comments and Suggestions for Authors

Congratulations and thank you to authors.

Comments on the Quality of English Language

English is so good

Author Response

Thank you again for your recommendations.

Reviewer 2 Report

Comments and Suggestions for Authors

Only some minor comments.

Reference 11, 15 and 17 are not correctly cited in the reference list. Please check.

Thnak you very much for your work
